# Modeling time-varying brain networks with a self-tuning optimized Kalman filter

D. Pascucci[1,2]*, M. Rubega[3,4], G. Plomp[1]

**1** Perceptual Networks Group, University of Fribourg, Fribourg, Switzerland, **2** Laboratory of Psychophysics, Brain Mind Institute, École Polytechnique Fédérale de Lausanne (EPFL), Lausanne, Switzerland, **3** Functional Brain Mapping Lab, Department of Fundamental Neurosciences, University of Geneva, Geneva, Switzerland, **4** Department of Neurosciences, University of Padova, Padova, Italy

* david.pascucci@epfl.ch

**Data Availability Statement:** Matlab and Python code for STOK, KF and the simulation framework are available on GitHub (https://github.com/PscDavid/dynet_toolbox; https://github.com/joanrue/pydynet).

## Abstract

Brain networks are complex dynamical systems in which directed interactions between different areas evolve at the sub-second scale of sensory, cognitive and motor processes. Due to the highly non-stationary nature of neural signals and their unknown noise components, however, modeling dynamic brain networks has remained one of the major challenges in contemporary neuroscience. Here, we present a new algorithm based on an innovative formulation of the Kalman filter that is optimized for tracking rapidly evolving patterns of directed functional connectivity under unknown noise conditions. The Self-Tuning Optimized Kalman filter (STOK) is a novel adaptive filter that embeds a self-tuning memory decay and a recursive regularization to guarantee high network tracking accuracy, temporal precision and robustness to noise. To validate the proposed algorithm, we performed an extensive comparison against the classical Kalman filter, in both realistic surrogate networks and real electroencephalography (EEG) data. In both simulations and real data, we show that the STOK filter estimates time-frequency patterns of directed connectivity with significantly superior performance. The advantages of the STOK filter were even clearer in real EEG data, where the algorithm recovered latent structures of dynamic connectivity from epicranial EEG recordings in rats and human visual evoked potentials, in excellent agreement with known physiology. These results establish the STOK filter as a powerful tool for modeling dynamic network structures in biological systems, with the potential to yield new insights into the rapid evolution of network states from which brain functions emerge.

## Author summary

During normal behavior, brains transition between functional network states several times per second. This allows humans to quickly read a sentence, and a frog to catch a fly. Understanding these fast network dynamics is fundamental to understanding how brains work, but up to now it has proven very difficult to model fast brain dynamics for various methodological reasons. To overcome these difficulties, we designed a new Kalman filter (STOK) by innovating on previous solutions from control theory and state-space modelling. We show that STOK accurately models fast network changes in simulations and real

**Funding:** This study was supported by the Swiss National Science Foundation grants to GP (PZ00P3_131731, PP00P1_157420 and CRSII5-170873), and to MR (CRSII5-170873). The funders had no role in study design, data collection and analysis, decision to publish, or preparation of the manuscript.

**Competing interests:** The authors have declared that no competing interests exist.

neural data, making it an essential new tool for modelling fast brain networks in the time and frequency domain.

This is a *PLOS Computational Biology* Methods paper.

## Introduction

Neural systems like the human brain exhibit highly dynamical patterns of neuronal interactions that evolve very quickly, at timescales of tens to hundreds of milliseconds. These temporal dynamics are fundamental for the coordination of large-scale functional networks at various oscillatory frequencies [1–4], both during rest [5–7] and in response to environmental events [8–11]. It is from such rapid and continuous reorganization of distributed neuronal interactions that sensory, motor and cognitive functions most likely arise [1–3]. To understand the workings of complex neural systems, it is therefore important to develop adequate models of their intrinsic dynamics that rely on the accurate estimation of time-varying functional connectivity patterns [12–16].

In the last decades, analysis of large-scale brain networks has successfully characterized the spatial layout and topology of functional connections [16–20], but their temporal dynamics have remained largely unexplored. The focus on network topologies instead of dynamics persists even though neural recordings with high temporal resolution are now readily available from advanced electrophysiology and neuroimaging techniques [21–25]. A major issue in modeling dynamic networks, particularly in the context of event-related responses, originates from the highly non-stationary nature of neural activity. Non-stationary signals pose severe modeling problems because of their unstable statistical properties, their time-varying spectral components and the multiple unknown sources of noise they contain [22,24,26]. To circumvent some of these problems, dynamic functional connectivity has been mostly estimated in a static or quasi-static sense, using for instance stationary measures applied to relatively long sliding windows [27–29]. Alternatively, model-based [30] and Markov Chain Monte Carlo methods [31,32] have been proposed for estimating dynamic connectivity under detailed a priori assumptions about the candidate generative processes and number of functional states [4,31,33,34]. Given the fast and flexible nature of brain activity, however, it is essential to move beyond static approximations of dynamical systems. This requires new algorithms that allow data-driven and large-scale exploration of functional brain networks at the sub-second scale of sensory, cognitive and motor processes.

Here, we present a new algorithm derived from control theory that is specifically designed to model dynamic changes in large-scale functional networks: the *Self-Tuning Optimized Kalman filter* (STOK). STOK belongs to the family of linear adaptive filters for estimating the temporal evolution of states in dynamical systems [35] and inherits the fundamental concepts of Kalman filtering [36]. The STOK filter combines three innovative solutions that set it apart from existing algorithms: 1) a simple least-squares minimization to recover latent and dynamic functional connectivity states through the adaptive estimation of time-varying multivariate autoregressive processes (tvMVAR), without any explicit approximation of unknown noise components; 2) a recursive regularization that guarantees robustness against noise while preventing overfitting; 3) a self-tuning memory decay that adapts tracking speed to time-varying

properties of the data, increasing the sensitivity to rapid transitions in connectivity states. Together, these three innovations define a new adaptive filter for modeling latent connectivity structures from fast-changing neural signals with unknown noise components.

We present an extensive validation of the proposed algorithm using a new simulation framework that mimics the realistic behavior of large-scale biological networks, and two data-sets of event-related potentials recorded in rodents and humans. In all quantitative tests and comparisons against the general linear Kalman filter (KF; [37,38]), we found that STOK shows unprecedented ability and precision in tracking the temporal dynamics of directed functional connectivity. The results establish the STOK filter as an effective tool for modeling large-scale dynamic functional networks from non-stationary time-series.

In the Methods section, we provide a detailed technical description of the STOK filter and of the limitations of the KF that it overcomes. Matlab and Python code for STOK, KF and the simulation framework are available on GitHub (https://github.com/PscDavid/dynet_toolbox; https://github.com/joanrue/pydynet).

## Results

### Simulations

A critical first step in the validation of the STOK filter was to evaluate whether the new filter successfully overcomes a well-known limitation of KF: the dependency of the filter's performance on a free parameter—the adaptation constant $c$ (see Methods), that determines the trade-off between tracking speed and smoothness of the estimates. As a proof of concept, we first compared a non-regularized version of the STOK filter against the KF in a simple two-nodes simulation. We used a bivariate AR(1) process (samples = 1000, Fs = 200 Hz, trials = 200) to generate signals with fixed univariate coefficients (A[1,1] = A[2,2] = 0.9) and a short sequence of causal influences from one node to the other (A[1,2] = 0.5). The results illustrate how KF performance depends critically on the fixed adaptation constant $c$. KF showed poor tracking ability at the lower bound ($c$ = 0.0001) and large noisy fluctuations at the higher bound ($c$ = 1) (Fig 1A). In contrast, the STOK filter thanks to its self-tuning $c$ automatically maintained a good level of performance by increasing its tracking speed at the on- and offsets of time-varying connections (Fig 1B). Computing the root-mean squared error against the imposed connection indicated that the optimal $c$ for KF lies at a point, within the two extremes, where the estimated coefficients from both filters are maximally correlated (Fig 1A, bottom plot). Note that, however, the determination of the optimal $c$ in real data is not straightforward and no objective or universal criteria are available [39]. This simulation shows that STOK can reach the peak performance of KF without prior selection of an optimal adaptation constant.

To statistically compare KF and STOK, we used a realistic framework with complex patterns of connectivity in the time and frequency domain. Our simulation framework allows for parametric variations of various signal aspects that can be critical in real neural data (see Methods; Simulation framework). As a first test, we evaluated the effect of regularization and the robustness of each filter against noise, comparing the performance of KF, STOK without regularization and STOK under different levels of SNR (0.1, 1, 3, 5, 10 dB). We used detection theory to compare simulated functional connectivity and estimated connectivity from KF and STOK, with area under the curve (AUC) as the performance metric (see Fig 2A and Methods). AUC values from 0.7 to 0.8 indicate fair performance, AUC from 0.8 to 0.9 indicate good performance [40]. A repeated measures ANOVA with factors Filter Type (KF, STOK without regularization, STOK) and signal-to-noise ratio (SNR), revealed a statistically significant interaction ($F(8, 232) = 68.54$, $p < 0.05$, $\eta_p^2 = 0.70$; Fig 2B). This effect demonstrated the

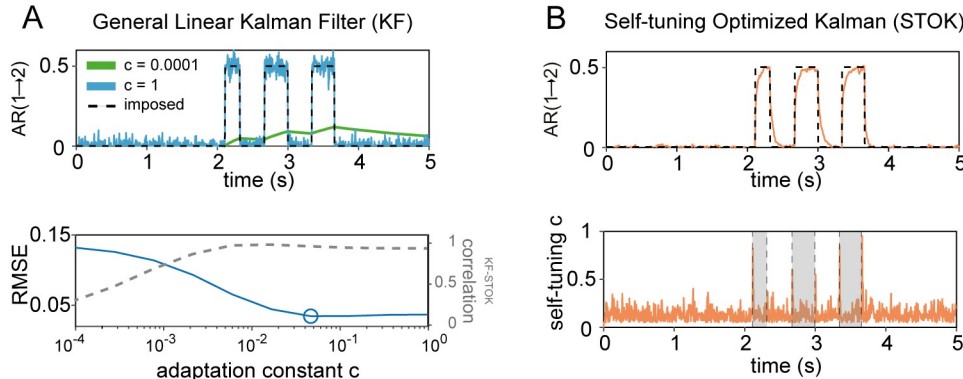

**Fig 1. Performance of KF and STOK on a simple simulated bivariate AR(1) process.** (A) Performance of the KF filter at recovering the imposed structure of AR coefficients (top panel; black dashed lines) under two extreme values of the adaptation constant ($c = 0.0001$, $c = 1$), highlighting the drastic variability of the estimates as a function of $c$: poor tracking performance is observed at the lowest (green line) and spurious noisy fluctuations at the highest $c$ (blue line). The optimal $c$ that minimizes the root-mean squared error (RMSE, blue line), lies at a point where KF and STOK performance are highly correlated (bottom panel; correlation shown by the grey dashed line). (B) Performance of the STOK filter, showing the high tracking ability and robustness to noise due to the self-tuning memory decay (top panel; orange line) which automatically increases tracking speed at relevant transition points between AR coefficient states (bottom panel; grey rectangles).

advantages of regularization: STOK showed better performance than KF across all noise levels (paired t-test, all $p < 0.05$), but the non-regularized STOK outperformed KF only for SNR larger than 0.1 (paired t-test, $p$ (SNR = 0.1) > 0.05; all other $p < 0.05$). Therefore, we kept regularization as a default component of STOK.

As a second test, we evaluated the robustness of KF and STOK against instantaneous linear mixing. Linear mixing, or spatial leakage [42,43], is an important issue when estimating functional connectivity from magneto- and electro-encephalographic data (M/EEG) because the multicollinearity and non-independence of multiple time-series can lead to spurious connectivity estimates [44,45]. Spatial leakage usually contaminates signals in nearby sources with a mixing profile that is maximal at around 10–20 mm of distance and fades out exponentially at around 40–60 mm [42–44]. To simulate linear mixing, we randomly assigned locations in a two-dimensional grid (150x150 mm) to each node of the surrogate networks ($n = 10$) and we convolved the signals, at each time point, with a spatial Gaussian point spread function (mixing kernel) of different standard deviations (10, 15, 20, 25, 30 mm). We then evaluated performance of the KF and STOK filters as a function of the mixing kernel width (Fig 2C). The results of a repeated measures ANOVA revealed an interaction between Filter Type and Mixing Kernel (F(4, 116) = 23.99, $p < 0.05$, $\eta_p^2 = 0.45$), with STOK outperforming KF for mixing functions up to 20 mm of width (paired t-test, $p < 0.05$). These results suggest that the STOK filter is preferred for small and intermediate mixing profiles that are observed in source imaging data [42,43] and in connectivity results [44]. For higher mixing levels, the filters showed indistinguishable but still fair performance.

Two additional tests compared the sensitivity of the STOK and KF to changes in autoregressive coefficients of various durations and sizes. In a first simulation, we imposed a single state of functional connectivity with variable duration (from 25 to 500 ms in ten linearly spaced steps). In the second test, we varied the maximal size of autoregressive coefficients, allowing their variation within different ranges (from 0.05 to 0.5, ten linearly spaced steps). These additional tests showed how STOK outperforms KF even for very transient and fast

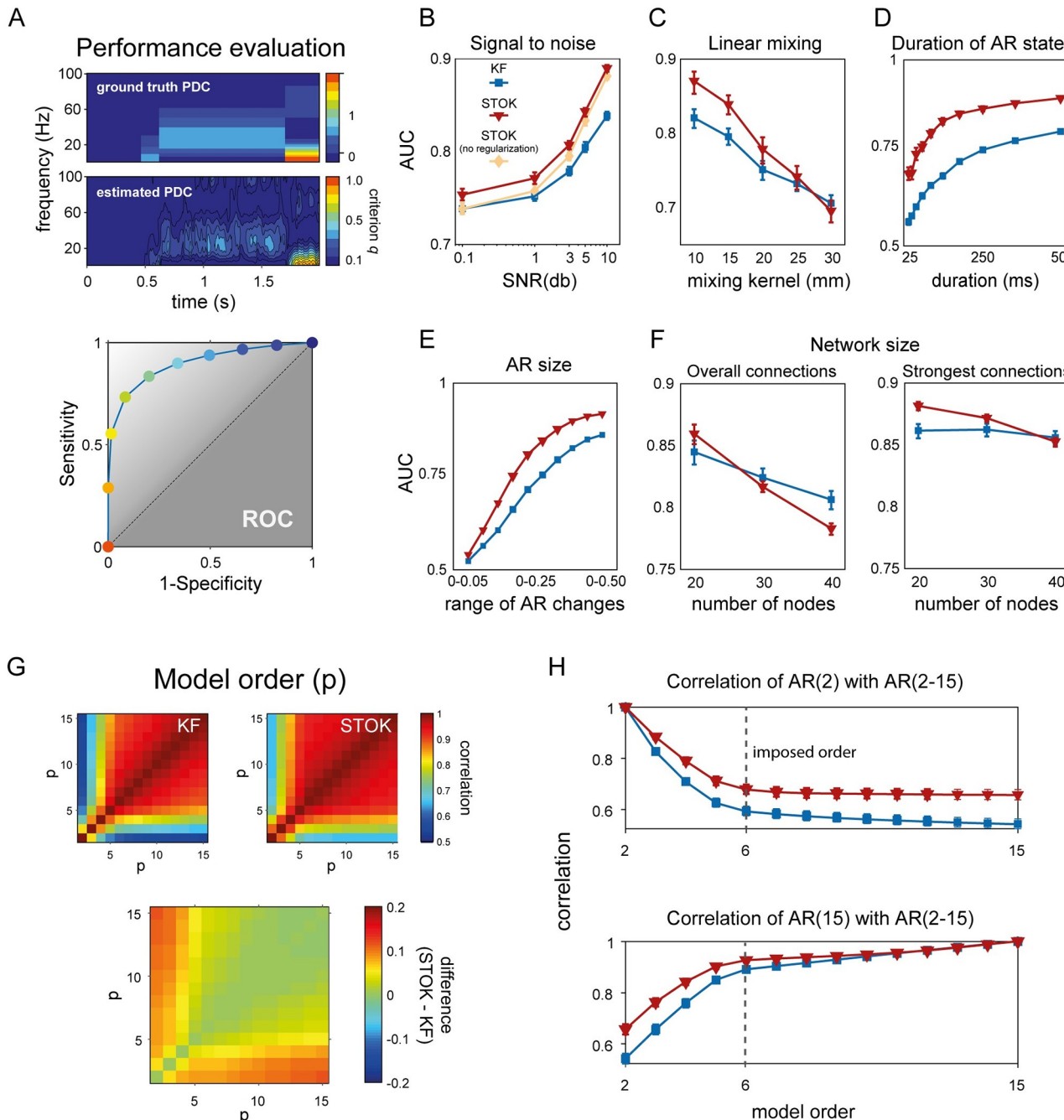

**Fig 2. Comparison of the KF and STOK filters under the realistic simulation framework.** (A) Method for evaluating the performance of KF and STOK against simulated data (ground truth). Ground truth PDC was binarized setting to 1 all connections larger than 0. Estimated Partial Directed Coherence (PDC) [41] was binarized using different criteria, based on the quantile discretization of the estimates (*criterion q*; top panel). Signal detection indexes were calculated for each criterion and the area under the curve (AUC) was used as performance measure. The color code of the dots in the ROC plot (bottom panel) reflects the different criteria and correspond to the colorbar for estimated PDC strength (top panel). (B) Comparison of KF, STOK without regularization and STOK as a function of different SNR, showing the overall larger AUC using STOK. Error bars reflect 95% confidence intervals. (C) AUC curves for KF and STOK as a function of linear mixing. (D-E) Performance of the two filters as a function of the duration (D) and size (E) of AR coefficients. (F) Performance of the two filters with increasing sample size: regularization favors strongest connections and sparse networks as the network size increases (right panel), reducing overall weakest connections (left panel). (G) Correlation matrices at varying model orders for KF and STOK (top two panels) and their difference (bottom panel, STOK minus KF). (H) Correlations extracted at specific orders ($p \in [2,15]$, with ground-truth model order = 6) showing the higher consistency of models estimated as $p$ changes with STOK compared with KF.

connectivity patterns (e.g., 25 ms of duration, Fig 2D; main effect of Filter Type: $F_{(1, 29)}$ = 4645.61, $p < 0.05$, $\eta_p^2 = 0.99$) while preserving higher tracking accuracy also when the change in the autoregressive coefficients was small (Fig 2E, main effect of Filter Type: $F_{(1, 29)}$ = 9631.31, $p < 0.05$, $\eta_p^2 = 0.99$).

Another critical aspect that determines the quality of the estimated parameters in the context of both multi-trial Kalman filtering [46,47] and ordinary least-squares solutions [48,49] is the number of parameters (e.g., nodes in the network). In general, to obtain robust parameter estimates and to avoid overfitting, a small ratio between parameters and number of trials is recommended (the one-in-ten rule of thumb) [50–52]. When this ratio is large (many parameters, few trials), the model is underdetermined and in this case regularization may help to prevent overfitting and to ensure that a unique solution is found [53]. Thus, increasing the number of nodes in our simulation allowed to test the behavior of the KF and STOK filters, as well as the effect of regularization, as the number of parameters in the model increased. We ran a set of simulations with fixed numbers of trials (n = 200) and increasing number of nodes (20, 30, 40). As expected, a repeated measures ANOVA with factors Filter Type and Number of Nodes revealed a significant interaction ($F_{(2, 58)}$ = 112.28, $p < 0.05$, $\eta_p^2 = 0.79$; Fig 2F) showing a decrease in performance with increasing number of nodes (main effect of the Number of Nodes, $F_{(2, 58)}$ = 63.68, $p < 0.05$, $\eta_p^2 = 0.68$). The interaction was due to a faster performance decrease for the STOK filter, which performed below KF levels for 30 and 40 nodes (paired t-test, both $p < 0.05$).

Regularization is designed to shrink weak coefficients toward zero and retain the strongest connections. We therefore examined whether the greater sensitivity to the number of nodes for the STOK filter was due to the diminishing of existing weak connections, by quantifying performance for the strongest connections only (magnitude above the 50% quantile). This reanalysis revealed an interaction between Filter Type and Number of Nodes ($F_{(2, 58)}$ = 45.73, $p < 0.05$, $\eta_p^2 = 0.61$; Fig 2F) in which the STOK outperformed KF for networks with 20 and 30 nodes (paired t-test, $p < 0.05$), while there was no significant difference for 40 nodes. Thus, for large-scale networks with a suboptimal ratio between the number of nodes and the number of trials, the regularized STOK filter provides a reliable sparse solution that accurately tracks the strongest dominant connections, while potentially preventing overfitting. Note that overall, however, performance was relatively good for both filters (AUC > 0.75).

As a final test in simulations, we investigated the robustness against variations in model order. The model order $p$ (Eq (2)) is a key free parameter in tvMVAR modelling that determines the amount of past information used to predict the present state influencing the quality and frequency resolution of the estimated auto-regressive coefficients [54]. Whereas previous work has shown that the multi-trial KF is relatively robust to variations in model order [46,55], we asked whether the innovations in STOK also make it more robust against changes in model order. We simulated data with an imposed order of $p = 6$ samples, and estimated PDC for both the STOK and the KF using a range of model orders from $p = 2$ to $p = 15$. As shown in Fig 2G and 2H, the correlation between PDC values obtained with different $p$ was overall higher for the STOK filter than for the KF. Particularly, the correlation was higher not only for $p \geq 6$, but also for smaller model orders, that usually lead to biased PDC estimates and poor frequency resolution.

In sum, the four tests in a realistic simulation framework showed that the STOK filter has superior performance, higher tracking accuracy and greater robustness to noise than the KF. STOK achieves these results without the need to set an adaptation constant, and with greater robustness to selecting a sub-optimal model order, two properties that are highly desirable when modeling real neural time-series. We next tested STOK performance in event-related EEG data recorded during whisker stimulation in rats, and during visual stimulation in humans.

## Somatosensory evoked potentials in rat

To compare STOK and KF along two objective performance criteria we used epicranial EEG recordings in rats from a unilateral whisker stimulation protocol [55–58]. Criterion I tests the ability to detect contralateral somatosensory cortex (cS1, electrode e4) as the main driver of evoked activity at short latencies after whisker stimulation (8–14 ms) in the gamma frequency band (40–90 Hz). Criterion II tests the identification of parietal and frontal areas (e2 and e6, respectively) as the main targets of cS1 (e4) in the gamma band, at early latencies [55,57]. To evaluate criterion I, we compared the summed outflow from cS1 with the largest summed outflow observed from the other nodes. To evaluate criterion II, we compared functional connectivity strengths from cS1 to e2 and e6 to that of the strongest connection directed to any of the other nodes. To determine the latencies at which KF and STOK are able to reliably identify cS1 as the main driver, and parietal-frontal cortex as their main targets, both criteria were evaluated at each timepoint around whisker stimulation (from -10 to +60 ms).

We evaluated performance on the two criteria using different sampling rates (1000 Hz, 500 Hz). The sampling rate determines the number of lags required to use a given model order in milliseconds, thus, it also determines the number of parameters in the model and the risk of overfitting [59]. Previous work has demonstrated that high sampling rates can have adverse effects on connectivity estimates [55,60] and that for these data the best connectivity performance for multi-trial KF is with downsampled data (500 Hz, [55]). For comparison, the model order for both methods and the adaptation constant for the KF were set to their previously reported optimal values ($p$ = 4 ms; $c$ = 0.02) [55].

At a sampling rate of 500 Hz, both the KF and the STOK filter revealed a peak in the summed gamma outflow from cS1 at early latencies from whisker simulation, Fig 3C. Both filters identified cS1 as the main driver (criterion I), by showing a significant increase of summed gamma outflow from cS1 at the expected latencies (bootstrap distribution of differences against the 2nd largest driver at each time point, $n$[bootstrap] = 10000, $p < 0.05$; Fig 3D). Similarly, for criterion II both methods identified e2 and e6 as the main targets of cS1 gamma influences, but the pattern was more restricted to the temporal window of interest in the STOK results (bootstrap distribution against the 2nd largest receiver at each time point; Fig 3E).

At the higher sampling rate of 1000 Hz, the STOK filter returned an almost identical pattern of outflow and good performance on both criteria (Fig 3F–3H). The KF, however, presented inconsistent outflows and poor performance on criterion I, failing to identify cS1 (e4) as the main driver of gamma activity. On criterion II, KF still performed well at high sampling rate (Fig 3H).

Overall, these benchmark results in real data show that STOK performs well on both performance criteria. In addition, it suggests that STOK has better specificity in the temporal domain, as compared to the KF results that presented interactions persisting at longer latencies without returning to baseline. Importantly, STOK performance was unaffected by the sampling rate used.

## Visual evoked potentials in human

As a final step, we compared the STOK and KF filters in real human EEG data from a motion discrimination task. The processing of coherent visual motion is known to induce characteristics time-frequency patterns of activity in cortical networks, with early selective responses occurring from 150 ms after stimulus onset [60–62] that likely originate in temporo-occipital regions (e.g., MT+/V5, V3a), and more pronounced responses from 250 ms on [63,64]. A hallmark of coherent motion processing is the induced broadband gamma activity from about 200

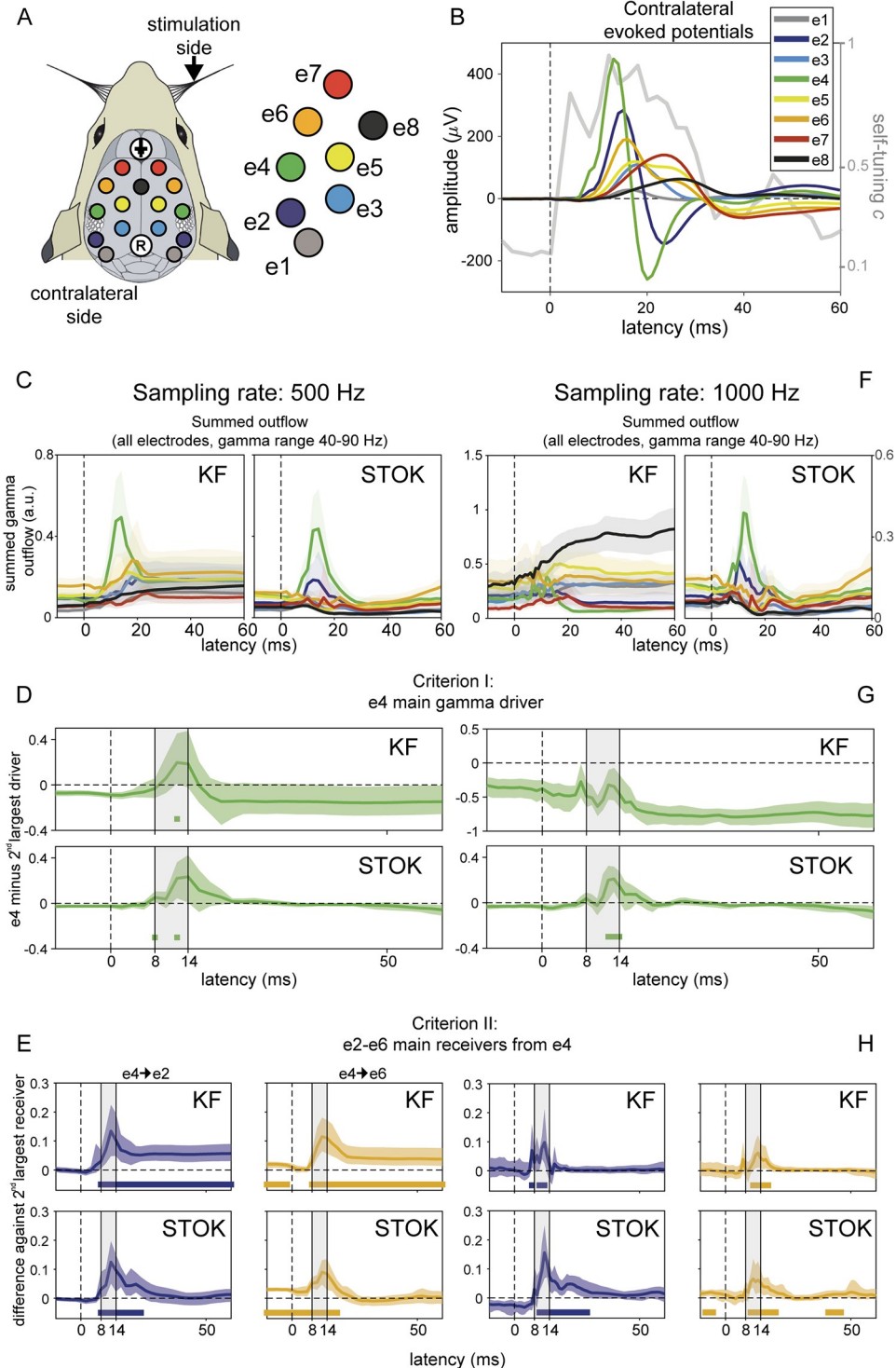

**Fig 3. Results in benchmark rat EEG.** (A) Layout of the multi-electrode grid used for recordings with the electrode and label codes used for all the plots. (B) Grand-average somatosensory evoked potentials at electrodes contralateral to stimulation ($n = 10$) showing the sequence of maximum voltage peaks, starting at e4 and propagating to e2-6. The gray line shows the evolution of the self-tuning memory parameter of the STOK filter. (C) Summed outflow in the gamma range (40–90 Hz) from all electrodes at the sampling rate of 500 Hz, revealing higher temporal precision with STOK filtering. (D-E) Criterion I and II: STOK and KF similarly identified e4 as the main driver at expected latencies (top panel), however, STOK recovered more temporally localized dynamics and evoked patterns in the total inflow of gamma activity from e4 to the two main targets e2-e6 (bottom panel). Colored squares at the bottom of each plot

indicate time points of significance after bootstrap statistics ($n$ = 10000, $p < 0.05$; see Results). (F-H) Same set of results using a sampling rate of 1000 Hz, revealing the compromised estimates of KF and the consistent and almost invariant results obtained with STOK.

ms onward [65–67], which is usually accompanied by event-related desynchronization in the alpha band [68].

To evaluate the performance of the STOK and KF filters at recovering known dynamics of coherent motion processing, we first compared the parametric power spectral density (PSD) obtained with each filter against the non-parametric PSD computed using Morlet wavelet convolution with linearly increasing number of wavelet cycles (from 3 to 15 cycles over the 1–100 Hz frequency range of interest; see Methods). As shown in Fig 4C, KF and STOK recovered the main expected dynamics in a qualitatively similar way as the non-parametric estimate. However, the STOK PSD showed significantly higher correlation with the non-parametric PSD as compared to the one obtained with the KF (Fig 4D, $r_{KF}$ = 0.53 ± 0.18; $r_{STOK}$ = 0.85 ± 0.05; $p < 0.05$; pairwise linear correlation between vectorized PSD). This shows that STOK produces more consistent PSD estimates across participants than KF. We note that both parametric methods appear to have higher temporal resolution than the non-parametric one, where temporal smoothing results from the trade-off between temporal and spectral resolution [69].

We next evaluated the overall time-frequency pattern of evoked functional connections obtained with STOK and KF. To this aim, we calculated PDC values from the tvMVAR coefficients estimated with the two filters and we averaged the results across nodes and hemifields. In this way, we obtained a global connectivity matrix of 16 cortical regions of interest (ROIs; see Methods, Human EEG) that summarized the evoked network dynamics in the time and frequency domain for each participant [70]. These matrices were then z-scored against a baseline period (from -100 to 0 ms with respect to stimulus onset) [71] and averaged across participants.

The resulting matrices of global event-related PDC changes revealed two critical differences between the STOK and the KF estimates. Firstly, STOK showed increased specificity in the temporal domain, as observed after collapsing across frequencies. While both filters showed an initial increase in global connectivity at early latencies (~110–120 ms post-stimulus), only the STOK filter, after a significantly faster recovery from the first peak (STOK vs. KF at 144–160 ms, $p < 0.05$), identified a second peak at critical latencies for motion processing (STOK vs. KF at 188–204 ms, $p < 0.05$) and a more pronounced decrease of global connectivity at a later stage (STOK vs. KF at 328–340 ms, $p < 0.05$; see Fig 4E). Interestingly, the second peak that STOK identified consisted of increased network activity in the high gamma band (70–90 Hz), and was due to due to increased outflow from motion- and vision-related ROIs that included areas MT+, V1 and FEF (see Fig 4E, bottom right). Secondly, the STOK filter showed increased specificity in the frequency domain. After collapsing the time dimension, STOK clearly identified decreased network activity at lower frequencies with a distinct peak in the higher alpha band (15 Hz), in agreement with the typical event-related alpha desynchronization, Fig 4C and 4E [68]. Contrarily, the network desynchronization profile estimated by the KF was less specific to the alpha range and more spread at lower and middle frequency bands (Fig 4E).

## Discussion

The non-stationarity nature of neuronal signals and their unknown noise components pose a severe challenge for tracking dynamic functional networks during active tasks and behavior.

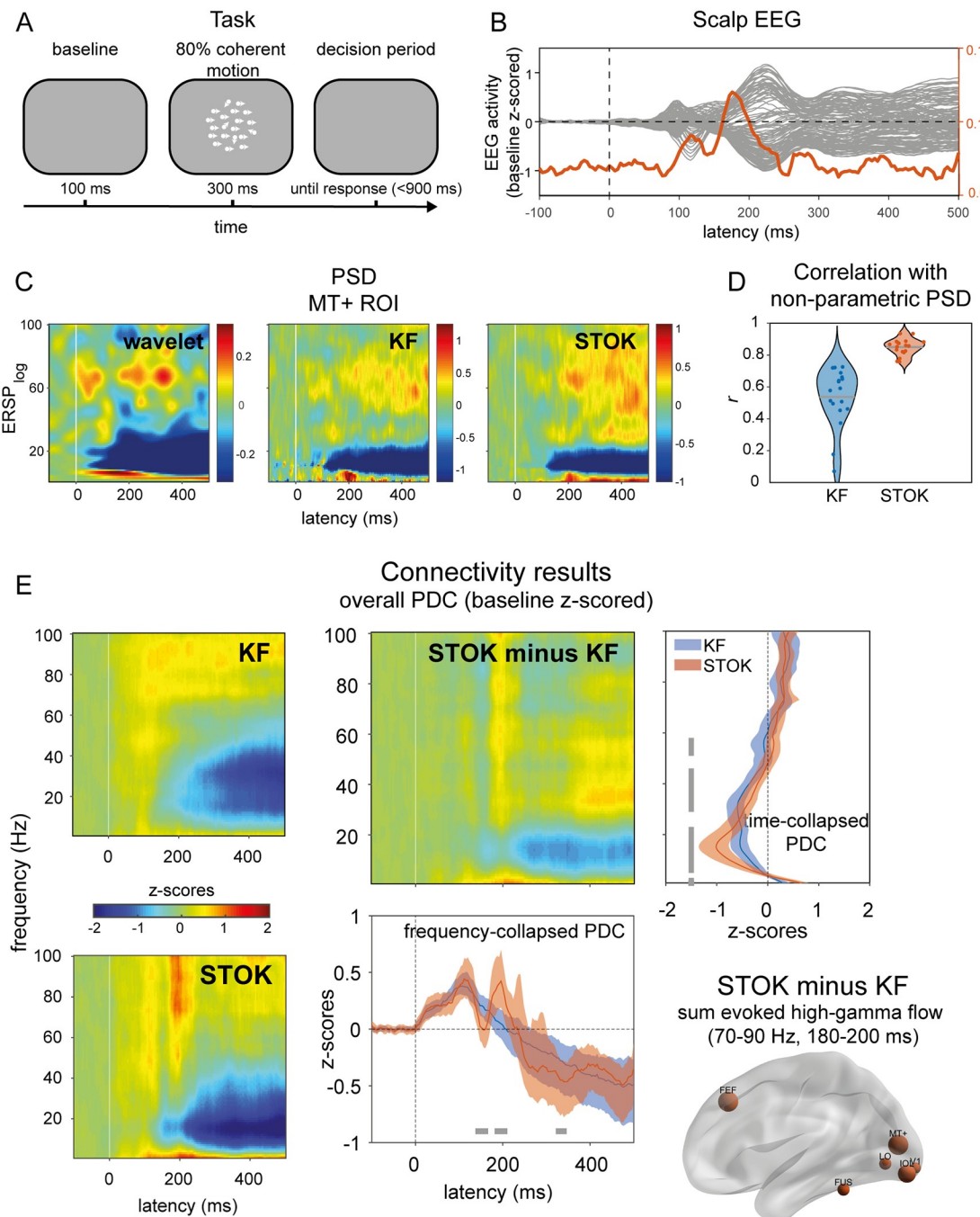

**Fig 4. Results in real human evoked potentials during visual motion discrimination.** (A) The visual motion discrimination paradigm presented during EEG recordings. Participants (n = 19) reported the presence of coherent motion in a briefly presented dot kinematogram (300 ms). (B) Shows grand-average event-related responses recorded at the scalp, with typical early (~100 ms) and late (~200 ms) components of visual processing. The orange line indicates the temporal dynamics detected by the self-tuning memory parameter *c*, that increases in anticipation of evident changes in the scalp signals. (C-D) Comparison of the non-parametric (wavelet) and parametric power spectrum densities (PSD) obtained with KF and STOK for one representative regions (MT+), with the violin plot showing the overall higher (and less variable) correlation between wavelet and STOK PSDs. (E) Global connectivity results from KF and STOK. Time-frequency plots show the results obtained with the two filters and their difference (STOK minus KF), graphically showing more evident dynamics obtained using the STOK filter. Line plots collapsing frequency and time highlight the statistical difference between STOK and KF results: STOK recovered multiple dynamic changes in overall connectivity patterns at physiologically plausible latencies (bottom plot) and characterized network desynchronization in the alpha range with higher precision (right-side plot). At frequencies (70–90 Hz) and latencies (180–200 ms) of interest for motion discrimination, the STOK revealed increased contribution to network activity (e.g., increased outflow) from visual regions, including MT+, and the frontal eye field (FEF; right-bottom plot).

In the present work, we have introduced and validated a new type of adaptive filter named the Self-Tuning Optimized Kalman filter (STOK). The STOK is optimized for tracking rapidly evolving patterns of directed connectivity in multivariate time-series of non-stationary signals, a challenge that makes most traditional algorithms inefficient. We designed the new adaptive filter with the goal to provide a tool for dynamic, frequency-resolved network analysis of multivariate neural recordings that is computationally efficient.

We validated the STOK filter using signal detection theory and an exhaustive battery of tests in simulated and real data. In a newly developed realistic simulation framework we showed that STOK outperforms the classical Kalman filter with better estimation accuracy in the time-frequency domain, higher tracking ability for varying SNR, and greater robustness to noise under signal mixing and simulated volume conduction effects [42,43]. In real data, STOK showed an unprecedented ability to recover physiologically plausible patterns of time-varying, frequency-resolved functional connectivity during whisker-evoked responses in rats and during visually evoked EEG responses in humans. It achieved such performance without any explicit approximation of unknown noise components and requiring only a single free parameter (the autoregressive model order $p$). Additional tests demonstrated that STOK performance was robust against variations of the model order and of the sampling rate, two aspects that are known to be critical for other algorithms [54–56]. These results validate STOK as a powerful new adaptive filter, optimized for uncovering network dynamics in multivariate sets of simultaneously recorded signals. This can have potentially broad applications in the field of systems and cognitive neuroscience, for the investigation of time-varying networks using evoked M/EEG response potentials, multi-unit activity, local field potentials (LFPs) and calcium imaging, or event-locked analyses like spike-triggered averages and traveling waves [21,23,24,72].

The accurate and robust performance of the STOK filter results from innovations based on existing engineering solutions. These innovations equip the filter with three important strengths: 1) it overcomes the problem of unknown design components in adaptive filtering [73], 2) it prevents overfitting and 3) it can track dynamical systems at variable speed [74,75]. Below, we discuss each of these three important properties.

To overcome the problem of unknown design components, the STOK filter extends an elegant least-squares simplification of the Kalman filter [73] to the case of multi-trial neuronal and physiological recordings. The advantages of this formulation, in which no explicit definition of the measurement and model uncertainty is required, are greatest when sources of uncertainty cannot be determined in advance, as is the case for recordings of neural activity. Recorded neural signals are usually contaminated by mixtures of noise that are hardly separable, including measurement noise, noise from the recording environment, biological artefacts and intrinsic fluctuations that are not pertinent to the process under investigation [22,24,26]. Approaches based on Kalman filtering can be drastically affected by suboptimal strategies for modelling noise components [35,73,76]. Various existing methods try to approximate unknown noise components directly from the data, based on innovations and residuals [77–79], covariance matching techniques [80], Bayesian, maximum-likelihood and correlation-based approaches [73,81], and other strategies adopted for neuroimaging [37,47,82]. However, in many cases the noise matrices estimated with such approximation methods may act as containers for unknown modelling errors [73], which leads to erroneous models and inadequate solutions [76,78]. To mitigate these risks, we adapted Nilsson's approach, which retains a simple and flexible formulation of the filter that is applicable to the case of multi-trial recordings and is agnostic to the measurement and process noise. An important caveat for a filter of this form, however, is that it is suboptimal: while avoiding potentially inaccurate approximations

of filter's components, overfitting and the inclusion of noise in the estimates becomes very likely.

In order to prevent overfitting, we introduced a regularization based on singular value smoothing [83]. Singular value smoothing, or damped SVD [84] retains information up to a given proportion of explained variance, reducing the effect of singular values below a given threshold (the filtering factor, Eq 20). Theoretically, how much to retain depends on the SNR and on the partitioning of variance among the main components of the data under investigation. For instance, lowering the amount of explained variance may result in connectivity estimates that are driven by only a few components. Whether this is desirable or problematic depends upon the hypothesis under consideration, and on the component structure in the data. At the other extreme, regularization can be avoided for very low-dimensional problems (e.g., bivariate analysis) or very high signal-to-noise ratio datasets. Following previous work, we set the filtering factor to retain 99% of the explained variance [85–88], and found that this threshold yields high and reliable performance for surrogate data of variable SNR, and for two sets of real EEG recordings.

As a least-squares regularization, the SVD smoothing also promotes sparse solutions by shrinking tvMVAR coefficients of irrelevant and redundant components toward zero. This feature helps to overcome the curse of dimensionality by favoring sparser connectivity patterns, a feature that is also expected in real functional brain networks, because of the sparse topology of the underlying structural links [89,90]. Promoting a certain degree of sparseness in functional connectivity has been also the objective of several recent works combining Granger causality and MVAR modelling with regularization procedures (e.g., $\ell_1$ or $\ell_2$ norm) [51,90–92]. Adding group-LASSO penalties, for instance, has been shown to improve the Kalman filter's sensitivity and the robustness of the estimates [93]. In its current form, STOK encourages sparse solutions by using a well-established technique for regularizing least-squares problems (SVD smoothing). The simplicity and flexibility of the least-squares reconstruction has the additional advantage that it becomes straightforward to implement other families of sparsity constraints, and to combine multiple constraints for the same estimate [90].

The third methodological innovation of the STOK filter is a self-tuning memory decay that automatically calibrates adaptation speed at each timepoint. The adaptation parameter is a critical factor in adaptive filtering that determines the trade-off between the filter's speed and the smoothness of the estimates [38,47]. Methods that use a fixed adaptation constant assume that the system under investigation has a constant memory decay. But this assumption is unlikely to hold for neural systems that show non-stationary dynamics and sequential states of variable duration [6,34,74,94]. To allow flexible tracking speed, adaptive filters with variable forgetting factors have been previously introduced, but these always require additional parameters that need to be chosen a priori, for instance to regulate the window length in which the forgetting factor is updated [75,95,96]. Here we developed a new solution to determine the memory of the system in a completely data-driven fashion, by updating the filter's speed using a window length of the model order $p$. At each time step, the residuals from independent past models of length $p$ are used to derive a recursive update of the filter, through the automatic regulation of an exponential running average factor $c$. By combining the self-tuning memory decay with SVD regularization, the filter can run at maximum speed without the risk of introducing noisy fluctuations in the estimates, a problem that we observed for the classical Kalman filter in both surrogate and real data (Figs 1 and 4). Unlike other algorithms, therefore, the STOK filter can accurately track phasic and rapid changes in connectivity patterns, such as those that may underlie sequential evoked components during tasks and event-related designs.

The temporal evolution of the memory parameter $c$ can potentially be used to indicate the presence of state transitions and stable states. When the model used to predict past segments

of data is no longer a good model for incoming data, the memory of the filter decreases and the algorithm learns more from new data than from previous predictions, indicating a potential state transition. Conversely, when past models keep predicting new data with comparable residuals, the filter presents longer memory and slower updates, suggesting a stable state. In this way, the temporal evolution of the memory decay provides information about time constants and transition points in the multivariate process under investigation, an additional indicator that can be used to quantify the temporal evolution of neural systems [4,16,94]. Measures of state stability and changes, for instance, have been previously used in topographic EEG analysis [25,97], and we expect these be related to temporal variations in system memory.

In its current form, the STOK filter is a multi-trial algorithm, leveraging regularities and correlations across trials under the assumption that multiple trials are coherent, temporally aligned realizations of the same process [38,98]. Because of this assumption, a proper realignment and time-locking of the events of interest is recommended before STOK filtering, particularly when differences in the duration and onset of the process under investigation are to be expected [99,100]. In principle, the algorithm can also be adapted for real-time tracking, continuous recordings and single-trial modelling, provided that the least-squares reconstruction at its core is not ill-conditioned. This can be achieved, for instance, by adding more of the past measurements to the observation equation (e.g., Eq (20) in Nilsson, 2006). In addition to proper time-locking of the events of interest, we also recommend the use of relatively long baseline segments to prevent artifacts due to the initial adaptation stage of the filter. Future work will address the suitability of the STOK for single-trial and real-time tracking with dedicated tests.

As a note of caution, STOK can be used to derive directed functional connectivity measures within the Granger causality framework which has well-known strengths and limitations [101–103]. As such, it estimates linear temporal dependencies and statistical relationships among multiple signals in a data-driven way, without a guaranteed mapping onto the underlying neuronal circuitry [26,104–106]. However, STOK provides a novel formulation that is well-suited for incorporating model-based or physiologically-derived information that could favor more biophysically plausible interpretations. Structural connectivity matrices, for instance, or models of cortical layers' communication, can be easily incorporated as priors for constraining the least-squares solution [107,108], thus allowing the estimation of dynamic functional connectivity on the backbone of a detailed biophysical model.

As evident from our tests on real stimulus-evoked EEG data, the STOK filter can recover key patterns of dynamic functional connectivity with high temporal and frequency resolution. This positions STOK to provide new insights into the fast dynamics of neural interactions that were previously unattainable due to methodological limitations. In the rat EEG data, for instance, STOK results indicated that gamma-band activity flows mainly from contralateral somatosensory cortex to neighboring regions in a restricted temporal window around the peak of evoked activity, followed by a global decrease of interactions that may underlie local post-excitatory inhibition and global desynchronization in the gamma range [109,110]. Before whisker stimulation, gamma-band influences from somatosensory cortex already showed increased functional connectivity with anterior, but not posterior, regions. Such detailed and temporally well-defined patterns of functional connections provide new valuable information for models of somatosensory processing in rats. Likewise, our results with human EEG recordings clearly indicated two critical windows of network interactions in the gamma range that emerged at plausible latencies of motion processing [65–67]. These interactions involved increased outflow from temporal-occipital regions, including MT+, and from the human homologue of the frontal eye field, providing a clear view on the network organization of motion processing.

As our results in human EEG suggest, STOK is also a promising tool for parametric time-varying power spectrum density estimation [111], as it is less affected by the choice of the model order compared to other parametric approaches [98]. Therefore, because of its ability to track fast temporal dynamics while maintaining high frequency specificity, STOK may be preferred for PSD analysis over non-parametric methods affected by the trade-off between temporal and frequency resolution [69].

To conclude, the STOK filter is a new tool for tvMVAR modeling non-stationary data with unknown noise components. It accurately characterizes event-related states, rapid network reconfigurations and frequency-specific dynamics at the sub-second timescale. STOK provides a powerful new tool in the quest of understanding fast functional network dynamics during sensory, motor and cognitive tasks [13,112,113], and can be widely applied in a variety of fields, such as systems-, network- and cognitive neuroscience.

## Methods

### Time-varying multivariate autoregressive modelling under the general linear Kalman Filter

Physiological time-series with multiple trials can be considered as a collection of realizations of the same multivariate stochastic process $Y_t$:

$$Y_t = \begin{bmatrix} y_{1,t}^{(1)} & \cdots & y_{d,t}^{(1)} \\ \vdots & \ddots & \vdots \\ y_{1,t}^{(N)} & \cdots & y_{d,t}^{(N)} \end{bmatrix} \qquad t = t_1, .., t_T \tag{1}$$

where $t$ refers to time, $T$ is the length of the time-series, $N$ the total number of trials and $d$ the dimension of the process (e.g., number of channels/electrodes). The dynamic behavior of $Y$ over time can be adequately described by a tvMVAR model of the general form:

$$Y_t = \sum_{k=1}^{p} A_{k,t} Y_{t-k} + \varepsilon_t \tag{2}$$

where $A_{k,t}$ are $[d \times d]$ matrices containing the model coefficients (AR matrices), $\varepsilon_t$ is the zero-mean white noise with covariance matrix $\Sigma_\varepsilon$ (also called the *innovation* process), and $p$ is the model order.

An efficient approach to derive the AR coefficients and the innovation covariance $\Sigma_\varepsilon$ in Eq 2 is the use of state-space models [35,37,38]. State-space models apply to problems with multivariate dynamic linear systems of both stationary and non-stationary stochastic variables [114] and can be used to reconstruct the set of linearly independent hidden variables that regulate the evolution of the system over time [26]. The general linear Kalman filter (KF) [36,38] is an estimator of a system state and covariance that has the following state-space representation:

$$x_t = \Phi_{t-1} x_{t-1} + \omega_{t-1} \tag{3}$$

$$z_t = H_t x_t + v_t \tag{4}$$

Eqs (3) and (4) are called the state or system equation and the observation or measurement equation, respectively. In Eq 3, the hidden state $x$ at time $t$ has a deterministic component given by the propagation of the previous state $x_{t-1}$ through a transition matrix $\Phi$, and a stochastic component given by the zero-mean white noise sequence $\omega$ of covariance $Q_t$. In Eq 4, the observed data $z$ at time $t$ are expressed as a linear combination of the state variable $x$ with

projection measurement matrix $H$, in the absence of noise. The term $v_t$ is a random white noise perturbation (zero mean, covariance $R_t$) corrupting the measurements.

To recursively estimate the hidden state $x$ at each time ($t = t_1,..,t_T$), the Kalman filter alternates between two steps, the prediction and the update step. In the prediction step, the state and the error covariance are extrapolated as:

$$\hat{x}_t^{(-)} = \Phi_{t-1}\hat{x}_{t-1}^{(+)} \tag{5}$$

$$P_t^{(-)} = \Phi_{t-1}P_{t-1}^{(+)}\Phi_{t-1}^T + Q_{t-1} \tag{6}$$

where $\hat{x}_t^{(-)}$ and $P_t^{(-)}$ are the *a priori* or predicted state and the error covariance at time $t$, based on the propagation of the previous estimated state and covariance $\hat{x}_{t-1}^{(+)}$ and $P_{t-1}^{(+)}$ through the transition matrix $\Phi$. The superscript $^T$ denotes matrix transposition. Note that Eq (6) contains an explicit term for the process noise covariance matrix $Q$.

In the update step, a posteriori estimates of the state and error covariance are refined according to:

$$K_t = P_t^{(-)}H_t^T(H_tP_t^{(-)}H_t^T + R_t)^{-1} \tag{7}$$

$$\hat{x}_t^{(+)} = \hat{x}_t^{(-)} + K_t(z_t - H_t\hat{x}_t^{(-)}) \tag{8}$$

$$P_t^{(+)} = (I - K_tH_t)P_t^{(-)} \tag{9}$$

where $I$ is the identity matrix, and $K_t$ is the Kalman Gain matrix reflecting the relationship between uncertainty in the prior estimate and uncertainty in the measurements (in more simple form, $k = \frac{\sigma^2_{estimate}}{\sigma^2_{estimate}+\sigma^2_{measurement}}$, with $\sigma^2$ = variance). The Kalman Gain thus quantifies the relative reliability of measurements and predictions and determines which one should be given more weight during the update step: if measurements are reliable, the measurement noise covariance $R_t$ is smaller and $K_t$ from Eq 7 will be larger; if measurements are noisy (larger $R_t$), $K_t$ will be smaller. The effect of $K_t$ on the updated a posteriori state estimate $\hat{x}_t^{(+)}$ is evident from Eq 8, where the updated state at time $t$ is a linear combination of the *a priori* state $\hat{x}_t^{(-)}$ and a weighted difference between the current measurements $z_t$ and the predicted measurement based on $\hat{x}_t^{(-)}$ (e.g., the residuals or measurement innovation term $(z_t - H_t\hat{x}_t^{(-)})$ on the right-hand side of Eq (8)). Thus, when the Kalman Gain increases following reliable measurements, the contribution of the measurement innovation will increase as well, and the a posteriori estimate $\hat{x}_t^{(+)}$ will contain more from actual measurements and less from previous predictions. Conversely, when the Kalman Gain decreases following noisy measurements, the a posteriori estimate $\hat{x}_t^{(+)}$ will be closer to the *a priori* predicted state $\hat{x}_t^{(-)}$. It is important to note that the Kalman Gain minimizes the trace of the prediction error covariance $P_t^{(+)}$ [35] and depends on the *innovation covariance* term $(H_tP_t^{(-)}H_t^T + R_t)^{-1}$ in Eq 7, which includes explicitly the measurement noise $R$ and the process noise covariance $Q$ from Eq 6. When both $w$ and $v$ are Gaussian with $w \sim N(0, R)$, $v \sim N(0, Q)$ and $E[w_tv_t^T] = 0$, and the design and noise matrices $H$, $\Phi$, $R$, and $Q$ are known, the state-space Kalman filter is the optimal linear adaptive filter [35].

In the context of physiological time-series, however, the optimal behavior of the Kalman filter is not assured and the algorithm requires some specific accommodations to account for: 1) the lack of known transition matrix $\Phi$ and measurement matrix $H$, and 2) the unknown covariance matrices $R$, and $Q$. To accommodate 1), the transition matrix $\Phi$ is usually replaced

by an identity matrix $I$ [37,38], which propagates the state $x$ from time $t − 1$ to $t$, such that the state in Eq (3) follows a first order random walk model [115]:

$$x_t = x_{t-1} + \omega_{t-1}. \tag{10}$$

The objective of the filter is to reconstruct the hidden tvMVAR process generating the observed physiological signals for each time $t$, which implies the following links between the state-space representation in Eqs (3) and (4) and the tvMVAR model in Eqs (1) and (2):

$$x_t = \begin{bmatrix} A_{1,t}^{(1)} \\ \vdots \\ A_{p,t}^{(N)} \end{bmatrix}, \quad z_t = Y_t \tag{11}$$

where $x_t$ has dimensions $[d^*p \times p]$ and $z$ $[N \times d]$ contains the measured signals at the current time $t$.

To establish the connection with the tvMVAR model, the measurement projection matrix $H$ is redefined as:

$$H_t = (Y_{t-1}, \ldots, Y_{t-p}) \tag{12}$$

such that measurement Eq (4) now expresses the observed data as a linear combination of the state $x_t$ and past measurements $H_t$ with additional perturbation $v_t$. This formulation suggests that the hidden state $x_t$ can be represented as a noise-contaminated least-squares reconstruction from present and past measurements:

$$x_t = H_t^{-1} z_t - v_t. \tag{13}$$

The second critical step in applying Kalman filtering to physiological data is the determination of the filter covariance matrices $R$, and $Q$. A widely used approach is to derive $R$ recursively from measurement innovations and to approximate $Q$ as a diagonal weight matrix that determines the rate of change of $P_t^{(-)}$ ([37,38,47], see also [116] for a list of alternative methods). With this approach, $\hat{R}$ is initialized as $I$ $[d \times d]$ and adaptively updated from the measurement innovations (the pre-update residuals) as:

$$\Sigma_r = \frac{(z_t - H_t \hat{x}_t^{(-)})^T (z_t - H_t \hat{x}_t^{(-)})}{N - 1}, \qquad \hat{R}_t = \hat{R}_{t-1} + c(\Sigma_r - \hat{R}_{t-1}) \tag{14}$$

where $\Sigma_r$ is the covariance of measurement innovations, $N$ is the total number of trials and $c$ $(0 \leq c \leq 1)$ is a constant across time that regulates the adaptation speed for $\hat{R}_t$ [38]. $\hat{R}_t$ is computed before the Kalman update to replace the unknown $R_t$ in the Kalman Gain with

$$K_t = P_t^{(-)} H_t^T (H_t P_t^{(-)} H_t^T + tr(\hat{R}_t) I_N)^{-1} \tag{15}$$

where $tr$ denotes the trace of a matrix and $I_N$ is the identity matrix $[N \times N]$. The other unknown process noise covariance $Q$, is replaced by a rate of change matrix $C^2 I_{[d \times p]}$ added to the diagonal of $P_t^{(-)}$ in Eq (6) [117]. The two constants $c$ and $C$ are usually selected as identical and determined *a priori* [38,118–120] or through cost functions that minimize residual errors [37,121,70]. In what follows we assume $c$ and $C$ to be identical and denote them as adaptation constant $c$.

The lack of a known transition matrix (Eq (5)) and the way that $R$ and $Q$ are approximated makes the adaptation constant $c$ the critical free parameter that determines the trade-off between fast adaptation and smoothness: a small $c$ value adds inertia to the system, reducing

the ability to track and to recover from dynamic changes in the true state while a large $c$ value increases the contribution of measurements to each update in Eq (7) and the uncertainty associated with $P_t^{(-)}$. Thus, setting $c$ too large yields highly variable estimates that fluctuate around the true state introducing disturbances to the estimated state, rather than filtering them out [122]. Although there exists no objective criterion to determine the optimal $c$ in real data [39], several optimization approaches are available [70,116,121], but they are not universal to all types of data [39]. Choosing $c$ *a priori* or based on previous findings is complicated by a further non-trivial aspect of the filter: the trace approximation of $R$ in Eq (15) $(tr(\hat{R}_t))$ implies that the system's dimensionality co-determines the uncertainty in measurements, the Kalman gain and the relative weight assigned to measurements. Thus, the effect of $c$ on the update depends on the number of signals considered. Moreover, $c$ is assumed stationary and constant for every time step $t$, but this assumption may not be warranted in the context of non-stationary neuronal time-series [74].

These critical aspects, along with the lack of an objective criterion for selecting $c$, increases the risk of erroneous models and suboptimal filtering of physiological data which complicates the validity of inferences and the generalization of findings.

## The STOK: Self-Tuning Optimized Kalman filter

The critical role of the adaptation constant $c$ when both $R$ and $Q$ are unknown motivated us to develop a new adaptive filter that presents the following properties: 1) It does not require any explicit knowledge of $R$ and $Q$ [73,123]; 2) It embeds a self-tuning factor that auto-calibrates the adaptation speed at each time step. Property 1) is achieved by extending the solution for Kalman filtering with unknown noise covariances proposed in Nilsson (2006, [73]) to the case of multi-trial time-series. According to Nilsson (2006, [73]), a reasonable tracking speed avoiding noise fluctuations can be achieved assuming the following relationship:

$$HPH^T \approx cR \tag{16}$$

that is, the error covariance matrix $P$, projected onto the measurement space, is a scaled version of the measurement noise covariance matrix $R$, with $c$ a scalar positive tuning factor (see [73] for a complete derivation). Assumption [16] allows a new formulation of the Kalman gain in Eq (7) as:

$$
\begin{aligned}
K_t &= P_t^{(-)} H_t^T (H_t P_t^{(-)} H_t^T + R_t)^{-1} \\
&= H_t^+ c R_t (c R_t + R_t)^{-1} \\
&= c H_t^+ (c+1)^{-1} = \frac{c}{1+c} H_t^+
\end{aligned}
\tag{17}
$$

where the apex + stands for the Moore-Penrose pseudoinverse. By substituting $K_t$ from Eq (17) in Eq (8), the new state update becomes:

$$
\begin{aligned}
\hat{x}_t^{(+)} &= \hat{x}_t^{(-)} + K_t (z_t - H_t \hat{x}_t^{(-)}) \\
&= \hat{x}_t^{(-)} + \frac{c}{1+c} H_t^+ (z_t - H_t \hat{x}_t^{(-)}) \\
&= \frac{\hat{x}_t^{(-)} + c H_t^+ z_t}{1+c}
\end{aligned}
\tag{18}
$$

in which the update of $\hat{x}_t^{(+)}$ is a weighted average of past predictions $\hat{x}_t^{(-)}$ and a least-squares reconstruction from recent measurements $H_t^+ z_t$. When $H_t$ is defined as in Eq (12), $H_t^+ z_t$ is

equivalent to finding the set of MVAR coefficients at each time $t$, by least-squares regression of the present signals $z_t$ on the past signals $H_t$, with multiple trials as observations.

The link with a least-squares problem was already suggested in Eq (13), however, by comparing Eq (13) with Eq (18), it is evident that the new state update does not incorporate any component of measurement noise $v_t$. This implies that in the presence of noisy measurements, the new filter might be susceptible to overfitting and sensitive to noise. To overcome this issue, we introduced regularization, a widely-used strategy to reduce model complexity and to prevent overfitting in the domain of least-squares problems [90,91,124]. More precisely, we employed a singular value decomposition (SVD)-based noise filtering with a standard form regularization [124,125] and a data-driven determination of the tuning parameter. Consider $\tilde{H}$, the SVD of the $N \times dp$ matrix $H$:

$$\tilde{H} = USV^T \tag{19}$$

where $U$ and $V$ are orthonormal matrices and $S$ is a $N \times N$ diagonal matrix of singular values in decreasing order. A regularized solution for the pseudoinverse $H^+$ used in Eq (18) can be derived from Eq (19) as

$$\tilde{H}^+ = V\Gamma_r^+ U^T, \quad \Gamma_r^+ = \begin{bmatrix} S_{1,1}/S_{1,1}^2+\lambda & \cdots & 0 \\ \vdots & \ddots & \vdots \\ 0 & \cdots & S_{N,N}/S_{N,N}^2+\lambda \end{bmatrix} \tag{20}$$

in which the diagonal elements of $\Gamma_r^+$ correspond to the diagonal of the inverse of $S$, subject to a smoothing filter that dampens the components lower than a tuning factor $\lambda$ [125]. To determine $\lambda$ in a completely data-driven fashion and to avoid excessive regularization, we use a variance-based criterion: At each time step, $\lambda$ takes on the value that allows to retain components that together explain at least 99% of the total variance in $H_t$. The 99% criterion is a canonical conservative threshold recommended in dimensionality reduction and noise filtering of physiological time-series [85–88], but the value of this threshold can in principle be tuned to the signal-to-noise ratio.

The second property that we introduced in the STOK filter is a self-tuning memory based on the adaptive calibration of the tuning factor $c$ in Eq (18). The single constant $c$ is a smoothing parameter in the exponential smoothing of the state $\hat{x}_t^{(+)}$ and determines the exponential decay of weights assigned to past predicted states, as they get older—the fading memory of the system. Whereas a fixed adaptation constant assumes a steady memory decay of the system, which could not be appropriate in modelling neuronal processes and dynamics [74], solutions for variable fading factors have been widely explored (see [126] for a comprehensive list), also in relation to intrinsic dynamics of physiological signals [127]. Here we propose a new method based on monitoring the proportional change in innovation residuals from consecutive segments of time, according to:

$$c_t = \min\left(b + \left[\frac{|tr(\hat{\Sigma}_\varepsilon^{new}) - tr(\hat{\Sigma}_\varepsilon^{old})|}{tr(\hat{\Sigma}_\varepsilon^{old})}\right], 1 - b\right) \tag{21}$$

where $b$ is a baseline constant ($b = 0.05$) that prevents the filter to perform at excessively slow tracking speed, such that $c \in (0.05, 0.95)$, and $tr(\hat{\Sigma}_\varepsilon^{new/old})$ is the trace of the estimated measurements innovation covariance for consecutive segments of data: *new* is a segment comprising samples from $t$ to $t - p$, and *old* is a segment from $t - (p + 1)$ to $t - 2p$. The use of successive

residuals to adjust variable fading factors, as well as the choice of segments or averaging windows to prevent spurious effects of instantaneous residuals, is common practice in adaptive filtering [75,128] but requires the selection of an additional parameter that specify the windows length. Here we set $p$—the model order—as the segments' length and compare residuals from two consecutive non-overlapping segments in order to adjust $c$ at each time $t$. The rationale behind this strategy is to avoid any additional parameter, considering the morel order (i.e., the amount of past information chosen to best predict the signals) as the optimal segment for extrapolating residuals. In addition, non-overlapping segments are used to monitor changes in residuals from independent sets of data. In other words, Eq (21) allows $c$ to increase as the residuals generated by the model in predicting new data increase with respect to an independent model from the immediate past: when the model is no longer capable of explaining incoming data, tracking speed increases and the memory of the system shortens.

## Partial Directed Coherence (PDC)

To compare STOK and KF using a time-frequency representation of directed connectivity, we computed the squared row-normalized Partial Directed Coherence ([41], PDC; [129]). PDC quantifies the direct influence from time-series $j$ to time-series $l$, after discounting the effect of all the other time-series. In its squared and row-normalized definition, PDC from $j$ to $l$ is a function of $A_{lj}$, obtained as:

$$\bar{\pi}_{lj}(f,t) = \frac{|\bar{A}_{lj}(f,t)|^2}{\sum_{m=1}^{d} |\bar{A}_{lm}(f,t)|^2} \qquad (22)$$

where $\bar{A}(f,t)$ is the frequency representation of the $A$ coefficients at time $t$, after the Z-transform:

$$\bar{A}(f,t) = \sum_{k=1}^{p} A_{k,t} z^{-k}, \quad z = e^{-i2\pi f} \qquad (23)$$

with $i$ as the imaginary unit. The square exponents in Eq (22) enhance the accuracy and stability of the estimates [129] while the denominator allows the normalization of outgoing connections by the inflows [57].

The parametric time-varying power spectral density of each time-series (PSD) can be estimated using the prediction error covariance matrix $\hat{\Sigma}_\varepsilon$ and the complex matrix in Eq (23), as:

$$PSD = B(f,t)\hat{\Sigma}_\varepsilon B(f,t)^* \qquad (24)$$

where $B(f,t)$ is the transfer function equal to the inverse of $\bar{A}(f,t)$, and $^*$ is the complex conjugate transpose. Since $\Sigma_\varepsilon$ is time invariant by definition, $\hat{\Sigma}_\varepsilon$ was estimated in both the KF and the STOK as the median measurements' innovation covariance (e.g., $\hat{R}_t$ in KF) across the last half of samples, in order to remove the effect of the initial filters' adaptation stage.

## Simulation framework

To systematically compare the STOK and KF performance against known ground truth, we developed a new Monte Carlo simulation framework that approximates properties of realistic brain networks, extending beyond classical approaches with restricted number of nodes and fixed connectivity patterns [28]. Signals were simulated according to a reduced AR(6) process in which coefficients of a AR(2) model were placed in the first two lags for diagonal elements, and at variable delays (up to 5 samples) for off-diagonal elements [130]. Surrogate networks were created assuming existing physical links among 60–80% of all possible connections [131]

and directed functional interactions were placed in a subset of existing links (50%) with variable time-frequency dynamics. Dominant oscillatory components in the low frequency range (e.g., 1–25 Hz) were generated by imposing positives values in the diagonal AR(2) coefficients of the simulated tvMVAR matrix [132]. Interactions at multiple frequencies were generated by randomly assigning both positive and negative values to the AR(2) coefficients outside the diagonal. The magnitude of AR coefficients was randomly determined (range: 0.1–0.5, in steps of 0.01) and off-diagonal coefficients were scaled by half magnitude. This range and scaling were chosen to match patterns observed in human EEG data.

To mimic dynamic changes in connectivity patterns, the structure and magnitude of off-diagonal AR coefficients varied across time, visiting three different regimes of randomly determined onset and transition times and with the only constrain to remain constant for at least 150 ms, approximating the duration of quasi-stationary and metastable functional brain states [6,133]. For each simulated regimen, the stochastic generation of AR coefficients was reiterated until the system reached asymptotic stability, i.e., satisfying the condition of real eigenvalues lower than zero.

Time-series for multiple trials (Fs = 200 Hz; duration = 2 s) were obtained by feeding the same tvMVAR process with generative zero-mean white noise of variance 1, and imposing a small degree of correlation ($r = 0.1 \pm 0.07$) in the generative noise across trials, reflecting the assumption that trials are realizations of the same process [38] and in line with the correlation among trials observed in the human EEG dataset. Except when specific parameters were varied, all simulations were done with 10 nodes, 200 trials and no additive noise. When additive noise was included in the simulation, the signal-to-noise ratio (SNR) was determined as the ratio between the squared amplitude of the signal and the squared amplitude of the additive noise.

To compare STOK and KF performance, we used the Receiving Operating Characteristic method (ROC) [40]. For each simulated network, we first obtained a target ground truth by calculating PDC values directly from the simulated tvMVAR matrices, for frequencies between 1 and 100 Hz. Separate PDC matrices were then computed from the AR coefficients estimated with the STOK and KF filters. The ground truth PDC values were binarized using a range of thresholds criteria (e.g., PDC > 0; or PDC > 0.5 quantile, see Fig 2A), defining zeros as signal absent and ones as signal present. Similarly, the estimated PDC values were binarized using a range of criteria at which connections were considered present or absent. The range of criteria consisted of twenty equally-spaced quantiles (from the 1$^{st}$ to the 99$^{th}$ quantile) from the distribution of each estimated PDC. Sensitivity and specificity indexes were then computed for each criterion against the ground truth PDC and used to derive the ROC curve. Finally, overall performance was quantified by the area under the ROC curve (AUC, see Fig 2A). This method has the advantage of being independent of the range of values in each estimated PDC and does not require any parametric or bootstrap procedure to determine statistically significant connections.

For each condition tested (see Results), we ran 30 realizations with different combination of parameters and the resulting AUC values were used in Analysis of Variance (ANOVA) and t-test statistical analysis using an alpha criterion of 0.05 for rejecting the null hypothesis.

## Benchmark rat EEG

These EEG data were previously recorded from a grid of 16 stainless steel electrodes placed directly on the skull bone of 10 young Wistar rats (P21; half males) during unilateral whisker stimulations under light isoflurane anesthesia (Fig 3A and 3B). All animal handling procedures were approved by the Office Vétérinaire Cantonal (Geneva, Switzerland) in accordance with

Swiss Federal Laws. Data were originally acquired at 2000 Hz and bandpass filtered online between 1 and 500 Hz. Additional details about the recording can be found elsewhere [57,58]. Data are freely available from https://osf.io/fd5ru.

The STOK filter and KF were applied to the entire network of 16 channels and used to derive PDC estimates. PDC results from the left and right stimulation were then combined within animals and only contralateral electrodes were further analyzed [57].

## Human EEG

These human EEG data were taken from an ongoing project aimed at investigating the connectivity patterns of functionally specialized areas during perceptual processing. Data were recorded at 2048 Hz with a 128-channel Biosemi Active Two EEG system (Biosemi, Amsterdam, The Netherlands) while nineteen participants (3 males, mean age = 23 ± 3.5) performed a coherent motion detection task in a dimly lit and electrically shielded room. Each trial started with a blank interval of 500 ms followed by a central dot kinematogram lasting 300 ms (dot field size = 8˚; mean dot luminance = 50%). In half of the trials, 80% of the dots were moving toward either the left or right, with the remaining 20% moving randomly. In the other half of trials, all dots were moving randomly. Participants had to report the presence of coherent motion by pressing one of two buttons of a response box (Fig 4A). After the participant's response, there was a random interval (from 600 to 900 ms) before the beginning of a new trial. There were four blocks of 150 trials each, for a total of 600 trials, (300 with coherent motion). Trials with coherent and random motion were interleaved randomly. Stimuli were generated using Psychopy [134] and presented on a VIEWPixx/3D display system (1920 ×1080 pixels, refresh rate of 100 Hz). All participants provided written informed consent before the experiment and had normal or corrected-to-normal vision. The experiment was approved by the local ethical committee.

EEG data were downsampled to 250 Hz (anti-aliasing filter: cut-off frequency = 112.5 Hz; transition bandwidth = 50 Hz) and detrended to remove slow fluctuations (<1 Hz) and linear trends [135]. The power line noise (50 Hz) was removed using the method of spectrum interpolation [136]. EEG epochs were then extracted from the continuous dataset and time-locked from -1500 to 1000 ms relative to stimulus onset. Noisy channels were identified before preprocessing and removed from the dataset (average proportion of channels removed across participants: 0.14 ± 0.06). Individual epochs containing non-stereotyped artifacts, peristimulus eye blinks and eye movements (occurring within ±500 ms from stimulus onset) were also identified by visual inspection and removed from further analysis (mean proportion of epochs removed across participants: 0.03 ± 0.03). Data were cleaned from remaining physiological artifacts (eye blinks, horizontal and vertical eye movements, muscle potentials) using a ICA decomposition (FastIca, eeglab; [137]). Bad ICA components were labelled by crossing the results of a machine-learning algorithm (MARA, Multiple Artifact Rejection Algorithm in eeglab) with the criterion of >90% of total variance explained. ICA selection and removal of the labelled components was performed manually (mean proportion of components removed: 0.07 ± 0.03). As a final pre-processing step, the excluded bad channels were interpolated using the nearest-neighbor spline method, data were re-referenced to the average reference and a global z-score transformation was applied to the entire dataset of each participant.

The LAURA algorithm implemented in Cartool [138] was used to compute the source reconstruction from available individual magnetic resonance imaging (MRI) data, applying the local spherical model with anatomical constraints (LSMAC) that constrains the solution space to the gray matter [138]. A parcellation of the cortex into 83 sub-regions was then obtained using the Connectome Mapper open-source pipeline [139] and the Desikan-Killiany

anatomical atlas [140]. Source activity was then extracted from 16 bilateral motion-related regions of interest (ROI) defined from the literature [141,142]. The ROIs were the pericalcarine cortex (V1), superior frontal sulcus (FEF), inferior parietal sulcus (IPS), cuneus (V3a), lateral occipital cortex (LOC), inferior medial occipital lobe (IOL), fusiform gyrus (FUS) and middle-temporal gyrus (MT+). Representative time-series for each ROI were obtained with the method of singular values decomposition [143]. Time-series were then orthogonalized to reduce spatial leakage effects using the innovation orthogonalization method [144] and estimating the mixing matrix from the residuals of a stationary MVAR model applied to a baseline pre-stimulus interval (from -200 to 0 ms). The optimal model order for each participant was also estimated from the stationary pre-stimulus MVAR model using the Akaike final prediction error criterion [145] (optimal $p = 11.9 \pm 1.2$). The optimal $c$ for KF was estimated using the Relative Error Variance criterion [70,121] (optimal $c = 0.0127$).

For the present work, we focused on EEG data in response to coherent motion only and we averaged the connectivity results from the left and right hemifield.

## Acknowledgments

We thank Mattia F. Pagnotta for helpful discussions, Joan Rué Queralt for translating the code to Python and Sebastien Tourbier for providing the brain parcellations based on the Connectome Mapper.

## Author Contributions

**Conceptualization:** D. Pascucci, M. Rubega, G. Plomp.

**Data curation:** D. Pascucci.

**Formal analysis:** D. Pascucci, M. Rubega.

**Funding acquisition:** G. Plomp.

**Investigation:** D. Pascucci, M. Rubega, G. Plomp.

**Methodology:** D. Pascucci, M. Rubega.

**Project administration:** G. Plomp.

**Resources:** G. Plomp.

**Software:** D. Pascucci, M. Rubega.

**Supervision:** G. Plomp.

**Validation:** D. Pascucci, M. Rubega.

**Visualization:** D. Pascucci.

**Writing – original draft:** D. Pascucci, G. Plomp.

**Writing – review & editing:** D. Pascucci, M. Rubega, G. Plomp.

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
