## [Decision Letter · Decision Letter 0]

24 Mar 2020

Dear Dr. Pascucci,

Thank you very much for submitting your manuscript "Modeling time-varying brain networks with a self-tuning optimized Kalman filter" for consideration at PLOS Computational Biology.

As with all papers reviewed by the journal, your manuscript was reviewed by members of the editorial board and by several independent reviewers. In light of the reviews (below this email), we would like to invite the resubmission of a significantly-revised version that takes into account the reviewers' comments.

We cannot make any decision about publication until we have seen the revised manuscript and your response to the reviewers' comments. Your revised manuscript is also likely to be sent to reviewers for further evaluation.

Sincerely,

Francesco P. Battaglia

Associate Editor

PLOS Computational Biology

Lyle Graham

Deputy Editor

PLOS Computational Biology

Reviewer's Responses to Questions

**Comments to the Authors:**

Reviewer #1: In this paper, the authors present a modification of the Kalman filter with an adaptive tuning parameter, intended to adjust to the non-stationarities in the smoothness of the signal. The method is applied to both synthetic and EEG data, with convincing results.

In my opinion, this is a very solid methodological paper, although has no major neuroscience/biology contribution.

I only have relatively minor comments:

- I can imagine the computational complexity could be an issue. The experimental data seems to be of small to moderate size (I could not find the duration of the data, please specify if missing). How would the method escalate to bigger data, both in terms of number of channels and number of time points?

- "In its current form, the STOK filter is a multi-trial algorithm, leveraging regularities and correlations across trials under the assumption that multiple trials are coherent, temporally aligned realizations of the same process". Could the authors elaborate on whether the method could be applied, or adapted, to continuous data? Also, previous work has hinted to possible differences in durations of stimulus processing between trials [1,2]. Could the authors comment on how the method would behave to differences in such timing? I think it could be interesting to comment on this, even if briefly.

[1] Stokes MG, Spaak E. 2016. The importance of single-trial analy-ses in Cognitive Neuroscience. Trends Cogn Sci. 20:483–486

[2] Vidaurre D, Myers N. Stokes MG, Nobre AC, Woolrich MW 2019 Temporally Unconstrained Decoding Reveals Consistent but Time-Varying Stages of Stimulus Processing. Cerebral Cortex Volume 29: 863–874

- As a general comment, it feels to me that the Discussion is too lenghty and methods-oriented for a journal focused on Computational Biology. I would suggest to make it a bit less dense.

- Importantly, the two links that the authors provide for the code don't work. Given the type of contribution that this paper makes, and the complexity of the implementation, I believe it's fundamental that the authors provide code in good shape. I have responded No to the question of whether all data was made available for this reason.

More minor:

- Please label the Figures more clearly. For instance, Fig 2F is confusing, what are exactly the two panels, and what is represented in the X-axis (given that the selected p is indicated in the title and the ground truth is 6)? Panels 3C don't have labels in the X-axis, etc.

- Fig 3F, what happens with KF at 1000Hz? I wouldn't say these look like "similar dynamics". In 3D, KF at 1000Hz looks also strange. Isn't it at odds with the statement "Previous work has demonstrated that downsampling can have adverse effects on connectivity estimates"?

Reviewer #2: review has been uploaded as an attachment

**Have all data underlying the figures and results presented in the manuscript been provided?**

Reviewer #1: No: Code links don't work

Reviewer #2: Yes

PLOS authors have the option to publish the peer review history of their article (what does this mean?). If published, this will include your full peer review and any attached files.

Reviewer #1: Yes: Diego Vidaurre

Reviewer #2: No
---

## [Decision Letter · Decision Letter 1]

3 Jul 2020

Dear Dr. Pascucci,

We are pleased to inform you that your manuscript 'Modeling time-varying brain networks with a self-tuning optimized Kalman filter' has been provisionally accepted for publication in PLOS Computational Biology.

Best regards,

Francesco P. Battaglia

Associate Editor

PLOS Computational Biology

Lyle Graham

Deputy Editor

PLOS Computational Biology

Reviewer's Responses to Questions

**Comments to the Authors:**

Reviewer #2: The authors have done a good job in responding to my previous critiques.

**Have all data underlying the figures and results presented in the manuscript been provided?**

Reviewer #2: Yes

PLOS authors have the option to publish the peer review history of their article (what does this mean?). If published, this will include your full peer review and any attached files.

Reviewer #2: No

---

## [Editor Report · Acceptance letter]

10 Aug 2020

PCOMPBIOL-D-19-02025R1 

Modeling time-varying brain networks with a self-tuning optimized Kalman filter

Dear Dr Pascucci,

I am pleased to inform you that your manuscript has been formally accepted for publication in PLOS Computational Biology. Your manuscript is now with our production department and you will be notified of the publication date in due course.

With kind regards,

Laura Mallard
